# Convergence of linear programming hierarchies for Gibbs states of spin systems

Hamza Fawzi[1] and Omar Fawzi[2]

[1]*DAMTP, University of Cambridge, United Kingdom*
[2]*Inria, ENS de Lyon, UCBL, LIP, France*

**Reviewed on OpenReview:** *https://openreview.net/forum?id=mc1dPxZsv3*

## Abstract

We consider the problem of computing expectation values of local functions under the Gibbs distribution of a spin system. In particular, we study two families of linear programming hierarchies for this problem. The first hierarchy imposes local spin flip equalities and has been considered in the bootstrap literature in high energy physics. For this hierarchy, we prove fast convergence under a spatial mixing (decay of correlations) condition. This condition is satisfied for example above the critical temperature for Ising models on a $d$-dimensional grid. The second hierarchy is based on a Markov chain having the Gibbs state as a fixed point and has been studied in the optimization literature and more recently in the bootstrap literature. For this hierarchy, we prove fast convergence provided the Markov chain mixes rapidly. Both hierarchies lead to an $\varepsilon$-approximation for local expectation values using a linear program of size quasi-polynomial in $n/\varepsilon$, where $n$ is the total number of sites, provided the interactions can be embedded in a $d$-dimensional grid with constant $d$. Compared to standard Monte Carlo methods, an advantage of this approach is that it always (i.e., for any system) outputs rigorous upper and lower bounds on the expectation value of interest, without needing an a priori analysis of the convergence speed.

## 1 Introduction

Consider a Gibbs distribution of the form

$$\mu(x) = \frac{e^{-H(x)}}{Z} \qquad (x \in \mathsf{S}) \tag{1}$$

where $H$ is the potential function, $Z = \sum_{x \in \mathsf{S}} e^{-H(x)}$, and $\mathsf{S}$ is the state space assumed finite. There are two fundamental computational questions that are associated to any such Gibbs distribution: (i) sampling from $\mu$, and (ii) computing expectation values, i.e., to compute $\mu(f) = \sum_{x \in \mathsf{S}} \mu(x)f(x)$ where $f : \mathsf{S} \to \mathbb{R}$ is a given function that we also call observable. In the vocabulary of graphical models, a Gibbs distribution is called a Markov random field and question (ii) is one of the typical computational inference problems studied in the literature on graphical models MacKay (2003); Wainwright et al. (2008).

An algorithm that generates samples from $\mu$ can be used to construct a randomized algorithm to compute expectation values. More precisely, if we have an algorithm that runs in time $T(\varepsilon)$ and produces a sample from a distribution $\nu$ whose total variation distance from $\mu$ is $\varepsilon$, then we have an algorithm that approximates $\mu(f)$ to within $\varepsilon + \delta$ with running time proportional to $T(\varepsilon)/\delta^2$ with probability, say, 2/3, assuming that $f$ takes values in the interval $[-1, 1]$.[1] Existing algorithms for (i), and hence (ii), are mostly based on Markov Chain Monte Carlo (MCMC). The idea is to run a Markov chain whose stationary distribution is $\mu$ until approximate convergence. The running time $T(\varepsilon)$ is then proportional to the mixing time of the Markov

---

[1]The $1/\delta^2$ factor comes from taking independent samples and ensuring the variance is $O(\delta^2)$.

chain. Despite its immense success, one drawback of most MCMC methods is the lack of a simple way to certify convergence.[2] In general, one needs to analyze the mixing time of the Markov chain beforehand, which is hard in many cases.

In this paper we focus on the problem of computing expectation values $\mu(f)$ and we are interested in *certified algorithms.* A certified algorithm will return, for each parameter $\ell$ quantities $\mathsf{p}_\ell^{\min}$ and $\mathsf{p}_\ell^{\max}$ such that $\mathsf{p}_\ell^{\min} \leq \mu(f) \leq \mathsf{p}_\ell^{\max}$. The parameter $\ell$ controls the running time of the algorithm, and as $\ell \to \infty$ we have $\mathsf{p}_\ell^{\max} - \mathsf{p}_\ell^{\min} \to 0$. Importantly, an a priori analysis of the convergence in $\ell$ is not needed to get certified bounds on $\mu(f)$. By computing $\mathsf{p}_\ell^{\min}$ and $\mathsf{p}_\ell^{\max}$ we have achieved an $\varepsilon$-approximation of the desired quantity with $\varepsilon = \mathsf{p}_\ell^{\max} - \mathsf{p}_\ell^{\min}$.

**Spin models**  We focus here on spin models where the state space $\mathsf{S}$ is the Cartesian product $\mathsf{S} = \Sigma^V$, where $\Sigma = \{-1, +1\}$ and $V$ is the vertex set of a graph $G = (V, E)$. The set $V$ is mostly assumed to be finite of size $n$, but we will highlight some of the results when they can equally be defined when $V$ is infinite. The potential function $H$ is of the form

$$H(x) = \beta \sum_{e=\{i,j\}\in E} h_e(x_i, x_j), \tag{2}$$

where $h_e : \Sigma^2 \to \mathbb{R}$ is a two-site interaction term which is assumed to be symmetric, i.e., $h_e(x_i, x_j) = h_e(x_j, x_i)$. For example, the ferromagnetic (resp. antiferromagnetic) Ising model corresponds to $h(x_i, x_j) = -x_i x_j$ (resp. $+x_i x_j$). The parameter $\beta \geq 0$ plays the role of an inverse-temperature. Note that when $\beta = 0$ then $\mu$ is the uniform distribution on $\mathsf{S}$, and when $\beta = +\infty$, $\mu$ is supported on the states $x \in \mathsf{S}$ that minimize $H$ (i.e., ground states).

**Contributions**  In this paper we study certified algorithms to approximate $\mu(f)$ when $f$ is a local function, i.e., depending only on variables in a set $B \subset V$ of small size. Note that the antiferromagnetic model encodes the MAX CUT problem for $\beta \to \infty$ and as such, computing $\mu(f)$ in general is hard. This means we have to make assumptions on $H$ in order to obtain provable convergence guarantees. We study two hierarchies of linear programs giving upper and lower bounds on $\mu(f)$. The two hierarchies are based on two different characterizations of the Gibbs measure:

1. The first one is based on imposing the so called DLR (Dobrushin-Lanford-Ruelle) equations Friedli & Velenik (2017) locally on a set $\Lambda \subseteq V$. The hierarchy is thus parametrized by sets $\Lambda$ satisfying $B \subseteq \Lambda \subseteq V$. For every such $\Lambda$, we define linear programs giving upper and lower bounds on the quantity of interest:
$$\mathrm{LP}_{\Lambda,\mathrm{DLR}}^{\min}(f) \leq \mu(f) \leq \mathrm{LP}_{\Lambda,\mathrm{DLR}}^{\max}(f). \tag{3}$$
Note that the interval $[\mathrm{LP}_{\Lambda,\mathrm{DLR}}^{\min}(f), \mathrm{LP}_{\Lambda,\mathrm{DLR}}^{\max}(f)]$ is monotone nonincreasing as a function of $\Lambda$ and for $\Lambda = V$, we have equality in (3). The linear programs parametrized by $\Lambda$ have a number of variables and constraints that are of order $2^{\Theta(|\Lambda|)}$. In Section 2, we prove that under a spatial mixing condition on the Gibbs distribution $\mu$, we have $|\mathrm{LP}_{\Lambda,\mathrm{DLR}}^{\max}(f) - \mathrm{LP}_{\Lambda,\mathrm{DLR}}^{\min}(f)| \leq \|f\|_\infty c_1 e^{-c_2 \mathrm{dist}(B,\Lambda^c)}$ for some constants $c_1, c_2 > 0$ (see Theorem 2.4) and $\Lambda^c = V \setminus \Lambda$. Here dist denotes the distance in the graph $G$. This implies that to achieve an approximation of $\varepsilon$, it suffices to choose the set $\Lambda$ so that $\mathrm{dist}(B, \Lambda^c)$ is sufficiently large while keeping $|\Lambda|$ as small as possible. The spatial mixing condition is satisfied for example for Ising models on a $d$-dimensional grid provided $\beta$ is below the critical inverse-temperature.

2. For the second hierarchy which is described in Section 3, we consider a Markov chain whose unique stationary distribution is $\mu$. Such Markov chains have been widely studied in the literature on spin systems, cf., Glauber dynamics Martinelli (1999). If $P$ is the Markov kernel describing the Markov chain, then stationarity of $\mu$ corresponds to the equations $\mu(Pg) = \mu(g)$ for all observables $g$. By imposing this condition on a set of local functions $g$ supported on some $\Lambda \subset V$, we arrive at a linear

---

[2]The technique of coupling from the past Propp & Wilson (1996) is a notable exception where exact convergence can be certified. It is in general more costly that standard MCMC.

programming relaxation on the stationary distributions of $P$. This allows us to obtain upper and lower bounds on the expectation value $\mu(f)$ for any local function $f$ supported on $B \subseteq \Lambda \subseteq V$

$$\text{LP}^{\min}_{\Lambda,\text{MC}}(f) \leq \mu(f) \leq \text{LP}^{\max}_{\Lambda,\text{MC}}(f). \tag{4}$$

The size of the linear programs defining $\text{LP}^{\min}_{\Lambda,\text{MC}}(f)$ and $\text{LP}^{\max}_{\Lambda,\text{MC}}(f)$ are also of the order of $2^{\Theta(|\Lambda|)}$, however they are slightly smaller than the one corresponding to the first hierarchy. In fact, as shown in Remark 3.2, under the assumption that $P$ is reversible, this hierarchy is provably weaker than the first one in the sense that for any $\Lambda$, $\text{LP}^{\min}_{\Lambda,\text{MC}}(f) \leq \text{LP}^{\min}_{\Lambda,\text{DLR}}(f)$ and $\text{LP}^{\max}_{\Lambda,\text{DLR}}(f) \leq \text{LP}^{\max}_{\Lambda,\text{MC}}(f)$. Nevertheless, we show in Theorem 3.3 that when the Markov chain has a mixing time of $O(n \log n)$, then we also have $|\text{LP}^{\max}_{\Lambda,\text{MC}}(f) - \text{LP}^{\min}_{\Lambda,\text{MC}}(f)| \leq \|f\|_\infty c_1' e^{-c_2' \text{dist}(B,\Lambda^c)}$ for some constants $c_1', c_2' > 0$.

The convergence result of the first hierarchy relies on the spatial mixing property, a property that solely depends on the Gibbs distribution. The convergence result of the second hierarchy however, relies on the fast temporal mixing of the Markov chain $P$ whose invariant distribution is $\mu$. It turns out that spatial mixing and fast temporal mixing of Glauber Markov chains are equivalent as shown in Stroock & Zegarlinski (1992). In fact, our proof of convergence for the second hierarchy uses a result from Dyer et al. (2004) providing a combinatorial argument for the fact that fast temporal mixing implies spatial mixing.

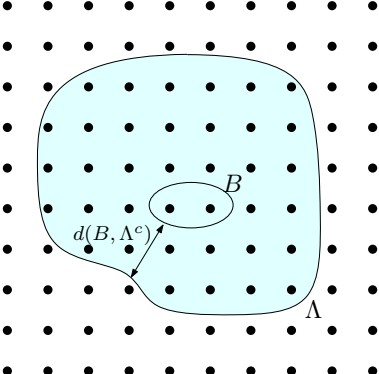

Figure 1: The linear programming hierarchies studied in this paper compute expectation values, with respect to the Gibbs distribution $\mu$, of observables $f : S \to \mathbb{R}$ with small support $\text{supp}(f) = B$. The hierarchies are indexed by subsets $\Lambda \supseteq B$, and convergence is expressed in the distance $\text{dist}(B, \Lambda^c)$.

**Related work**  The motivation for this paper came from recent work in the theoretical physics literature studying the bootstrap method, and in particular the papers Cho et al. (2022); Cho & Sun (2023) where the two hierarchies considered here have been studied numerically, and where asymptotic convergence in the case of infinite systems have been proven. The numerical results of Cho et al. (2022); Cho & Sun (2023) show a performance that is comparable to Markov chain Monte Carlo, in particular for systems that are away from criticality. A natural question left open by these works is to prove fast convergence of such certified methods for systems that are far from criticality. Our goal in this paper is to address this question. We note that the use of linear programming to compute moments of invariant distributions of general Markov chains goes back to the late 1990s, see e.g., Hernández-Lerma & Lasserre (1998) and (Hernández-Lerma & Lasserre, 2012, Chapter 12). The problems considered in this paper are also closely related to the literature on graphical models and variational inference, see e.g., Wainwright et al. (2008) and references therein. We note in particular the paper Risteski (2016) which analyzes a convex programming hierarchy to estimate the free energy of Ising models in the case of dense graphs. Another recent paper closely connected to our paper is Bach (2024) which proposes new algorithms based on sums of squares and quantum entropies to estimate log-partition functions. One difference between our paper and the two previously-cited papers is that we consider the problem of computing expectation values of observables directly, rather than the log-partition function. Finally we note that for quantum spin systems, certified algorithms for computing expectation values were recently proposed in Fawzi et al. (2024).

**Notations** For a subset $\Lambda \subseteq V$, the external boundary of $\Lambda$ is defined as

$$\partial^{ex}\Lambda = \{j \in \Lambda^c : \exists i \in \Lambda, \{i,j\} \in E\} \tag{5}$$

and we let $\bar{\Lambda} = \Lambda \cup \partial^{ex}\Lambda$. For a spin configuration $x \in \mathsf{S} = \Sigma^V$ and $i \in V$ we let $x^i$ be the configuration where the $i$'th spin is flipped. Given a function $f : \mathsf{S} \to \mathbb{R}$, $i \in V$ and $x \in \mathsf{S}$ we let

$$\nabla_i f(x) = f(x^i) - f(x). \tag{6}$$

The support of a function $f : \mathsf{S} \to \mathbb{R}$ is the set of spins that $f$ depends on, i.e.,

$$\mathrm{supp}(f) = \{i \in V : \nabla_i f \neq 0\}. \tag{7}$$

Note that if $\mathrm{supp}(f) \subseteq \Lambda$, then $\mu(f) = \mu_\Lambda(f)$ where $\mu_\Lambda$ is the marginal of $\mu$ on the sites $\Lambda$ and we slightly abused notation by using the same notation for $f$ and its restriction to $\Sigma^\Lambda$. Finally, given an arbitrary finite set $X$ we let $D(X) = \{p : X \to \mathbb{R}_+ : \sum_{x \in X} p(x) = 1\}$ be the set of probability distributions on $X$.

## 2  Linear programming relaxation via the DLR equations

Note that the Gibbs distribution $\mu$ defined in (1) satisfies the following identity, for any $x \in \mathsf{S}$ and $i \in V$:

$$\frac{\mu(x)}{\mu(x^i)} = e^{\nabla_i H(x)}.$$

In fact, $\mu$ is the unique probability distribution satisfying these equalities. Now fix $\Lambda \subseteq V$ and assume that $i \in \Lambda$. By locality of the Hamiltonian, the support of $\nabla_i H$ is included in $\bar{\Lambda}$ and so it is easy to see that the equation above is also true for the marginal $\mu_{\bar{\Lambda}}$ on $\bar{\Lambda} = \Lambda \cup \partial^{ex}\Lambda$:

$$\mu_{\bar{\Lambda}}(x) = \mu_{\bar{\Lambda}}(x^i) e^{\nabla_i H(x)} \qquad \forall x \in \Sigma^{\bar{\Lambda}} \ \forall i \in \Lambda. \tag{8}$$

These are known as the spin-flip equations, or the DLR equations. This leads us to define

$$\mathrm{LP}_{\Lambda,\mathrm{DLR}} = \left\{ \nu \in D(\Sigma^{\bar{\Lambda}}) : \nu(x) = \nu(x^i) e^{\nabla_i H(x)} \quad \forall x \in \Sigma^{\bar{\Lambda}}, i \in \Lambda \right\}. \tag{9}$$

By construction, the marginal $\mu_{\bar{\Lambda}}$ belongs to $\mathrm{LP}_{\Lambda,\mathrm{DLR}}$. Given a function $f : \mathsf{S} \to \mathbb{R}$ supported on $\Lambda$, we define the following upper and lower bounds on the expectation value $\mu(f)$:

$$\mathrm{LP}_{\Lambda,\mathrm{DLR}}^{\max}(f) = \max\{\nu(f) : \nu \in \mathrm{LP}_{\Lambda,\mathrm{DLR}}\} \qquad \mathrm{LP}_{\Lambda,\mathrm{DLR}}^{\min}(f) = \min\{\nu(f) : \nu \in \mathrm{LP}_{\Lambda,\mathrm{DLR}}\}. \tag{10}$$

Note that $\mathrm{LP}_{\Lambda,\mathrm{DLR}}^{\max}(f)$ and $\mathrm{LP}_{\Lambda,\mathrm{DLR}}^{\min}(f)$ are linear programs having $|\Sigma|^{|\bar{\Lambda}|}$ variables and $|\Lambda||\Sigma|^{|\bar{\Lambda}|}$ equality constraints.

**Relation to local Gibbs states** The following proposition gives an alternative characterization of the polyhedron $\mathrm{LP}_{\Lambda,\mathrm{DLR}}$. Before stating the proposition, we need to introduce the following important definition.

**Definition 2.1.** Let $\Lambda \subset V$ and $\eta \in \Sigma^{\partial^{ex}\Lambda}$. We define the *local Gibbs state* on $\Lambda$ with boundary conditions $\eta$ to be

$$\mu_\Lambda^\eta(x) = \frac{1}{Z_\Lambda^\eta} \exp(-H_\Lambda^\eta(x)) \qquad \forall x \in \Sigma^\Lambda$$

where

$$H_\Lambda^\eta(x) = \sum_{\substack{\{i,j\} \in E \\ i,j \in \Lambda}} h_{ij}(x_i, x_j) + \sum_{\substack{\{i,j\} \in E \\ i \in \Lambda, j \in \partial^{ex}\Lambda}} h_{ij}(x_i, \eta_j)$$

is the Hamiltonian truncated to $\Lambda$ with boundary conditions $\eta$ and $Z_\Lambda^\eta = \sum_{x \in \Sigma^\Lambda} \exp(-H_\Lambda^\eta(x))$.

**Proposition 2.2.** *If $\nu \in \mathrm{LP}_{\Lambda,\mathrm{DLR}}$ then there is a probability distribution $p(\eta)$ on $\Sigma^{\partial^{ex}\Lambda}$ such that for any $f : \mathsf{S} \to \mathbb{R}$ supported on $\Lambda$ we have*

$$\nu(f) = \sum_{\eta \in \Sigma^{\partial^{ex}\Lambda}} p(\eta)\mu_\Lambda^\eta(f).$$

*Proof.* Let $\nu \in \mathrm{LP}_{\Lambda,\mathrm{DLR}}$. Fix $\eta \in \Sigma^{\partial^{ex}\Lambda}$ an arbitrary spin configuration on $\partial^{ex}\Lambda$. The constraint in (9) tells us that for any $x \in \Sigma^\Lambda$

$$\frac{\nu(x,\eta)}{\nu(x^i,\eta)} = \exp(\nabla_i H_\Lambda^\eta(x)) \qquad \forall i \in \Lambda.$$

This means that the conditional distribution of $\nu$ on $\Lambda$ given that the boundary values are $\eta$ is precisely equal to the local Gibbs distribution $\mu_\Lambda^\eta$. Hence if we let $p(\eta) = \sum_{x \in \Sigma^\Lambda} \nu(x,\eta)$ be the marginal of $\nu$ on the boundary, we get the desired equality. $\qquad \square$

**Convergence** The main result in this section shows that if $\mu$ has spatial mixing, then the linear programming-based upper and lower bounds will converge exponentially fast (in $\mathrm{dist}(\mathrm{supp}(f),\Lambda^c)$) to $\mu(f)$. We adopt the following definition of spatial mixing, which corresponds to weak spatial mixing in (Martinelli, 1999, Definitions 2.3).

**Definition 2.3.** We say that a probability measure $\mu$ on $\Sigma^V$ has the *spatial mixing* property if there exist constants $C_1, C_2 > 0$ such that the following holds: for any two subsets $B \subseteq \Lambda \subseteq V$ and any function $f$ supported on $B$

$$\sup_{\eta,\tau \in \Sigma^{\partial^{ex}\Lambda}} |\mu_\Lambda^\eta(f) - \mu_\Lambda^\tau(f)| \le \|f\|_\infty C_1 \sum_{i \in B, j \in \partial^{ex}\Lambda} \exp(-C_2 \mathrm{dist}(i,j)). \tag{11}$$

For simplicity of expressions, we will be using the following implied inequality

$$\sup_{\eta,\tau \in \Sigma^{\partial^{ex}\Lambda}} |\mu_\Lambda^\eta(f) - \mu_\Lambda^\tau(f)| \le \|f\|_\infty C_1 |B| |\partial^{ex}\Lambda| \exp(-C_2 \mathrm{dist}(B,\Lambda^c)). \tag{12}$$

**Infinite systems** If the set of sites $V$ is infinite, e.g., the Ising model on the grid $V = \mathbb{Z}^d$, defining a Gibbs distribution is more subtle. The most standard definition is via the DLR equations: we say that a probability measure $\mu$ on the set of configurations $\{-1,+1\}^V$ is a Gibbs state if for all finite $\Lambda \subset V$, the marginal of $\mu$ on $\bar{\Lambda}$ satisfies (8). It turns out that for infinite systems, the Gibbs distribution might not be unique. Note that Definition 2.3 can be used as is for infinite systems as it never refers to Gibbs distributions of the whole system but rather to the distributions $\mu_\Lambda^\eta$ which only depend on the Hamiltonian restricted to $\bar{\Lambda}$. In fact, this condition is known to hold for Ising models on the infinite grid $\mathbb{Z}^d$ provided $\beta$ is below the corresponding critical temperature (Martinelli, 1999, page 102). Note that when spatial mixing holds, it is simple to see that there is a unique Gibbs distribution that we denote $\mu$.

**Theorem 2.4.** *Assume that the Gibbs distribution $\mu$ has the spatial mixing property of Definition 2.3. Then there are constants $C_1, C_2$ such that for any $B \subseteq \Lambda \subseteq V$ and any $f : S \to \mathbb{R}$ supported on $B$:*

$$\mathrm{LP}_{\Lambda,\mathrm{DLR}}^{\max}(f) - \mathrm{LP}_{\Lambda,\mathrm{DLR}}^{\min}(f) \le \|f\|_\infty C_1 |B| |\partial^{ex}\Lambda| \exp(-C_2 \mathrm{dist}(B,\Lambda^c)) . \tag{13}$$

*As a result, if the graph is a $d$-dimensional grid, $\mu(f)$ can be approximated with additive error $\varepsilon$ by choosing $\Lambda$ to be a ball of radius $\Theta(\log(|B|/\varepsilon))$ around the sites in $B$ which leads to a linear program of size $\exp(O(\log^d(|B|/\varepsilon)))$.*

Note that this theorem holds even when $V$ is infinite.

*Proof.* For any $\nu \in \mathrm{LP}_{\Lambda,\mathrm{DLR}}$, we know (from Proposition 2.2) that there is a probability distribution $p(\eta)$ on $\eta \in \Sigma^{\partial^{ex}\Lambda}$ such that

$$\nu(f) = \sum_\eta p(\eta)\mu_\Lambda^\eta(f).$$

As such for any $f$ supported on $B \subseteq \Lambda$, and any $\nu, \nu' \in \mathrm{LP}_{\Lambda,\mathrm{DLR}}$, we have

$$|\nu(f) - \nu'(f)| \le \sup_{\eta,\tau \in \Sigma^{\partial^{ex}\Lambda}} |\mu_\Lambda^\eta(f) - \mu_\Lambda^\tau(f)| \le \|f\|_\infty C_1 |B| |\partial^{ex}\Lambda| \exp(-C_2 \mathrm{dist}(B,\Lambda^c)),$$

which proves the desired statement. $\qquad \square$

## 3  Linear programming relaxation via a Markov chain

We now consider a second linear programming hierarchy to compute expectation values of Gibbs state $\mu$. This hierarchy is based on the characterization of the Gibbs distribution $\mu$ as the unique stationary measure of some local Markov chain. We start by introducing these local Markov chains which are also known as *Glauber dynamics*. Recall that a Markov chain on $\mathsf{S}$ is described via the transition probabilities $P_{x,x'}$ of moving from $x \in \mathsf{S}$ to $x' \in \mathsf{S}$.

**Definition 3.1.** We say that a Markov chain on $\mathsf{S}$ with transition matrix $P = (P_{x,x'})_{x,x' \in \mathsf{S}}$ is *local* if it satisfies the following:

1. Single-site update: $P_{x,x'} = 0$ if $x'$ differs from $x$ on more than one site

2. Locality: For any $x \in \mathsf{S}$ and $i \in V$, $P_{x,x^i}$ depends only on spins $x_j$ for $j$ in the neighborhood of $i$

3. Boundedness: $P_{x,x'} \leq \frac{1}{n}$ for all $x \neq x' \in \mathsf{S}$.

We can interpret such a Markov chain as follows: at each time step, a site $i \in V$ is chosen uniformly at random, and the $i$'th spin is flipped with a certain probability $c(i,x) \in [0,1]$. In this case the probability transition matrix is given by: $P_{x,x^i} = c(i,x)/n$, $P_{x,x} = 1 - \sum_{i \in V} c(i,x)/n$, and $P_{x,x'} = 0$ otherwise. To satisfy the assumption of locality the function $c(i,x)$ should only depend on the spins $x_j$ where $j$ is in the neighborhood of $i$.

We will be interested in local Markov chains whose stationary distribution is the Gibbs distribution $\mu$. A sufficient condition for $\mu$ to be stationary for $P$ is that $P$ is reversible for $\mu$, i.e., $\mu(x)P_{x,x'} = \mu(x')P_{x',x}$. For local Markov chains, this reversibility condition simplifies to:

$$P_{x,x^i}\mu(x) = P_{x^i,x}\mu(x^i) \qquad \forall i \in V,\ x \in \mathsf{S}$$

which, using the definition of $\mu$ is equivalent to

$$P_{x,x^i} = P_{x^i,x}e^{-\nabla_i H(x)} \tag{14}$$

where

$$\nabla_i H(x) = H(x^i) - H(x).$$

The simplest concrete example of a Markov chain satisfying the above assumptions is the heat-bath dynamics, where a given site $i \in V$ is flipped with probability

$$c(i,x) = \frac{\exp(-H(x^i))}{\exp(-H(x)) + \exp(-H(x^i))} = \frac{e^{-\nabla_i H(x)}}{1 + e^{-\nabla_i H(x)}}. \tag{15}$$

Note that $\nabla_i H(x)$ only depends on the spins $x_j$ for sites $j$ that are neighbors of $i$, and so $c(i,x)$ is local as desired.

**Observables**  Assume $(X_t)_{t \geq 0}$ is a Markov chain with transition matrix $P$. If $f : \Sigma^V \to \mathbb{R}$ is an observable, then $Pf(x) = \sum_{x' \in \mathsf{S}} P_{x,x'}f(x')$ is the expectation value of $f(X_1)$ conditioned on $X_0 = x$. More generally, $(P^t f)(x)$ is the expectation of $f(X_t)$ conditioned on $X_0 = x$. Using the locality of $P$, observe that

$$Pf(x) = P_{x,x}f(x) + \sum_{i \in V} P_{x,x^i}f(x^i) = f(x) + \sum_{i \in V} P_{x,x^i}\nabla_i f(x), \tag{16}$$

where we used the fact that $P_{x,x} = 1 - \sum_{i \in V} P_{x,x^i}$. Note that $\nabla_i f(x) = 0$ for $i \notin \operatorname{supp}(f)$. Since $P_{x,x^i}$ only depends on the spins $x_j$ in the neighborhood of $i$, the above shows that $\operatorname{supp}(Pf) \subseteq \overline{\operatorname{supp}(f)}$, where we recall that $\overline{\operatorname{supp}(f)}$ denotes the union of $\operatorname{supp}(f)$ with its external boundary.

**LP hierarchy based on Glauber dynamics** Assume $P$ is a local Markov chain for which $\mu$ is an invariant distribution, i.e., $\mu P = \mu$. The latter equality can be written equivalently as: $\mu(Pg) = \mu(g)$ for all observables $g : \mathsf{S} \to \mathbb{R}$. If we restrict $g$ to be supported on a certain $\Lambda \subseteq V$, then note that $\mu(Pg)$ and $\mu(g)$ only depend on the marginal of $\mu$ on $\bar{\Lambda}$. Given a subset $\Lambda \subseteq V$ we can thus consider the following set of distributions $\nu$ on $\bar{\Lambda}$:

$$\mathrm{LP}_{\Lambda,\mathrm{MC}} = \Big\{ \nu \in D(\Sigma^{\bar{\Lambda}}) : \begin{array}{ll} \nu(1) = 1 & \text{(Normalization)} \\ \nu(h) \geq 0 \quad \forall h \geq 0, \ \mathrm{supp}(h) \subseteq \bar{\Lambda} & \text{(Positivity)} \\ \nu(Pg) = \nu(g) \quad \forall g : \mathrm{supp}(g) \subseteq \Lambda & \text{(Stationarity)} \end{array} \Big\}. \tag{17}$$

By construction, the marginal of $\mu$ on $\bar{\Lambda}$ belongs to $\mathrm{LP}_{\Lambda,\mathrm{MC}}$. Hence if $f$ is any observable supported on $B \subseteq \bar{\Lambda}$ we can obtain lower and upper bounds on $\mu(f)$ by solving the following linear programs:

$$\mathrm{LP}_{\Lambda,\mathrm{MC}}^{\min}(f) = \min_{\nu}\{\nu(f) : \nu \in \mathrm{LP}_{\Lambda,\mathrm{MC}}\} \qquad \mathrm{LP}_{\Lambda,\mathrm{MC}}^{\max}(f) = \max_{\nu}\{\nu(f) : \nu \in \mathrm{LP}_{\Lambda,\mathrm{MC}}\}.$$

These are linear programs in $|\Sigma|^{|\bar{\Lambda}|}$ variables and at most $|\Sigma|^{|\Lambda|}$ equality constraints.

**Remark 3.2** (Relation with DLR equations). We show in this remark that if $P$ is reversible, then $\mathrm{LP}_{\Lambda,\mathrm{DLR}} \subseteq \mathrm{LP}_{\Lambda,\mathrm{MC}}$. Let us rewrite the stationarity condition of (17) more explicitly. If $\mathrm{supp}(g) \subseteq \Lambda$, and $\nu$ a distribution on $\Sigma^{\bar{\Lambda}}$, one can check that

$$\begin{aligned}
\nu(Pg) - \nu(g) &= \sum_{x \in \Sigma^{\bar{\Lambda}}} \nu(x)((Pg)(x) - g(x)) \\
&= \sum_{x \in \Sigma^{\bar{\Lambda}}} \nu(x) \sum_{i \in V} P_{x,x^i} \nabla_i g(x) \\
&= \sum_{x \in \Sigma^{\bar{\Lambda}}} \sum_{i \in \Lambda} \nu(x) P_{x,x^i} \nabla_i g(x) \\
&= \sum_{x \in \Sigma^{\bar{\Lambda}}} \sum_{i \in \Lambda} (\nu(x^i) P_{x^i,x} - \nu(x) P_{x,x^i}) g(x),
\end{aligned}$$

where in the last equality we used the fact that $\sum_{x \in \Sigma^{\bar{\Lambda}}} f(x) \nabla_i g(x) = \sum_{x \in \Sigma^{\bar{\Lambda}}} \nabla_i f(x) g(x)$. By taking $g$ to be an indicator function of $\sigma_\Lambda \in \Sigma^\Lambda$ we see that the equation $\nu(Pg) = \nu(g)$ in the definition of $\mathrm{LP}_{\Lambda,\mathrm{MC}}$ is equivalent to the following linear equation in $\nu$:

$$\sum_{\substack{x \in \Sigma^{\bar{\Lambda}} \\ x_\Lambda = \sigma_\Lambda}} \sum_{i \in \Lambda} \nu(x^i) P_{x^i,x} - \nu(x) P_{x,x^i} = 0. \tag{18}$$

The outer summation over $x$ is effectively a summation over spin values on the boundary $\partial^{ex}\Lambda = \bar{\Lambda} \setminus \Lambda$.

Now using the assumption that $P$ is reversible with respect to $\mu$, i.e., that (14) is satisfied, condition (18) for all $\sigma \in \Sigma^\Lambda$ simplifies to

$$\sum_{\substack{x \in \Sigma^{\bar{\Lambda}} \\ x_\Lambda = \sigma_\Lambda}} \sum_{i \in \Lambda} P_{x^i,x}\Big(\nu(x^i) - \nu(x) e^{-\nabla_i H(x)}\Big) = 0 \qquad \forall \sigma \in \Sigma^\Lambda. \tag{19}$$

We see that these equation are implied by the DLR equations (9); more precisely we see that the DLR equations impose that each term $\nu(x^i) - \nu(x) e^{-\nabla_i H(x)}$ of the sum is equal to zero. In other words, we have $\mathrm{LP}_{\Lambda,\mathrm{DLR}} \subseteq \mathrm{LP}_{\Lambda,\mathrm{MC}}$.

**Convergence** We are now ready to state our main convergence theorem.

**Theorem 3.3.** *Assume $G$ is a graph of bounded degree $\Delta$ with $n = |V|$ vertices. Let $P$ be a local Markov Chain according to Definition 3.1 and assume that there are constants $c_1, c_2 > 0$ such that for all $t \geq 0$,*

$$\sup_{\nu} \|\nu P^t - \mu\|_1 \leq c_1 n (1 - c_2/n)^t. \tag{20}$$

*Let $f : \Sigma^V \to \mathbb{R}$ supported on $B \subset V$. Then there are constants $C_1, C_2 > 0$ that depend only on $\Delta$, $|B|$, and $c_1, c_2$ such that the following holds: for any $\Lambda \supseteq B$*

$$\mathrm{LP}^{\max}_{\Lambda,\mathrm{MC}}(f) - \mathrm{LP}^{\min}_{\Lambda,\mathrm{MC}}(f) \leq C_1 n \|f\|_\infty e^{-C_2 \mathrm{dist}(B,\Lambda^c)}.$$

**Remark 3.4.** To get an accuracy $\varepsilon$, the above theorem means that we need to take $\Lambda$ such that $ne^{-\mathrm{dist}(B,\Lambda^c)} = O(\varepsilon)$, i.e., $\mathrm{dist}(B, \Lambda^c) = \Omega(\log(n/\varepsilon))$. If the graph is a $d$-dimensional grid, and $\Lambda$ is chosen to be a ball of radius $r = \Omega(\log(n/\varepsilon))$ centered at $B$ then $|\Lambda| = O(r^d)$ and so the LP has size $2^{r^d}$. So the running time for accuracy $\varepsilon$ is $O(\exp(C \log^d(n/\varepsilon)))$.

**Remark 3.5.** The condition (20) is known to hold for the heat bath dynamics defined in (15) on Ising models at sufficiently small inverse-temperature $\beta$. More specifically, it holds for any Ising model on a graph with degree bounded by $\Delta$ provided $\Delta \tanh(\beta) < 1$ (Levin & Peres, 2017, Theorem 15.1)

We first need the following important lemma about the speed of propagation of information in local Markov chains. We have seen that if $f$ is supported on $B \subset V$, then $P^t f$ is supported on $B^t := \{i \in V : \mathrm{dist}(i, B) \leq t\}$. The lemma shows that $P^t f$ can actually be well approximated by a function supported on $B^{ct/n}$ for some constant $c > 0$. This result is well-known, see e.g., (Martinelli, 1999, Lemma 3.2). Here, we adapt the proof of (Dyer et al., 2004, Lemma 3.1).

**Lemma 3.6.** *Assume the graph $G$ has $n = |V|$ nodes and is of maximum degree $\Delta > 1$, and let $P$ be a local Markov chain on $G$. Let $f : \Sigma^V \to \mathbb{R}$ be an observable supported on $B$, and let $\Lambda \supset B$. Call $r = \mathrm{dist}(B, \Lambda^c)$. Assume $x, x'$ are two spin configurations such that $x_i = x'_i$ for all $i \in \Lambda$. Then for any integer $t \geq 0$,*

$$|P^t f(x) - P^t f(x')| \leq \|f\|_\infty c e^{vt/n-r}.$$

*where $c = 8|B|$ and $v = e^2(\Delta - 1) > 0$.*

*Proof.* This is a direct consequence of (Dyer et al., 2004, Lemma 3.1) on the speed of propagation of information of the Glauber dynamics. We reproduce the argument here for convenience. Let $(X_t)_{t \geq 0}$ and $(Y_t)_{t \geq 0}$ be two copies of the Glauber dynamics with initial configurations respectively $x$ and $x'$ coupled as follows. At each time step $t \geq 1$, the vertex $i_t$ is chosen uniformly at random in $V$ and $U \in [0, 1]$ is a uniform random variable. Then $X_{t+1}$ is obtained by flipping the spin $i_t$ of $X_t$ when $U \leq c(i_t, X_t)$ and otherwise $X_{t+1} = X_t$. Similarly, $Y_{t+1}$ is obtained by flipping the spin $i_t$ of $Y_t$ when $U \leq c(i_t, Y_t)$. Note that if $X_t$ and $Y_t$ coincide on the neighborhood of the vertex $i_t$, then $X_{t+1}[i_t] = Y_{t+1}[i_t]$, where we use the notation $X[i]$ to denote the $i$-th spin of the configuration $X$. More generally for a set $B \subseteq V$, we let $X[B]$ be the string of spins in the set $B$. From the interpretation mentioned above, setting $X_0 = x$ and $Y_0 = x'$, we have

$$\begin{aligned}
|P^t f(x) - P^t f(x')| &= |\mathbb{E}[f(X_t) - f(Y_t)]| \\
&\leq \Pr[X_t[B] \neq Y_t[B]] 2\|f\|_\infty \\
&\leq \sum_{j \in B} \Pr[X_t[j] \neq Y_t[j]] 2\|f\|_\infty.
\end{aligned}$$

Note that in order to have $X_t[j] \neq Y_t[j]$, there has to be a path of disagreements from $\Lambda^c$ to $j$, i.e., $v_0, v_1, \ldots, v_l$ with $v_0 \in \Lambda^c$ and $v_l = j$. Note that for a fixed path $v_0, \ldots, v_l$, the probability that there exist times $0 < t_1 < t_2 < \cdots < t_l \leq t$ such that $i_{t_\alpha} = v_\alpha$ for all $\alpha$ is at most $\binom{t}{l}\left(\frac{1}{n}\right)^l$. The number of simple paths of length $l$ going from $\Lambda^c$ to $j$ is at most $\Delta(\Delta - 1)^{l-1}$. Recall that $\mathrm{dist}(\Lambda^c, B) = r$ which implies that $l \geq r$. Thus,

$$\begin{aligned}
\Pr[X_t[j] \neq Y_t[j]] &\leq \sum_{l=r}^{+\infty} \Delta(\Delta - 1)^{l-1} \binom{t}{l}\left(\frac{1}{n}\right)^l \\
&\leq \frac{\Delta}{\Delta - 1} \sum_{l=r}^{\infty} \left(\frac{e(\Delta - 1)(t/n)}{l}\right)^l.
\end{aligned}$$

If $r \geq e^2(\Delta - 1)(t/n)$, we bound

$$\Pr[X_t[j] \neq Y_t[j]] \leq 4\left(\frac{e(\Delta - 1)(t/n)}{r}\right)^r.$$

In this case, this leads to

$$|P^t f(x) - P^t f(x')| \leq 2\|f\|_\infty 4|B|\left(\frac{e(\Delta - 1)(t/n)}{r}\right)^r$$
$$\leq 8\|f\|_\infty|B|\exp(-r\log r) \leq \|f\|_\infty c\exp(vt/n - r).$$

In the case $r < e^2(\Delta - 1)(t/n)$, then we simply use the bound

$$|P^t f(x) - P^t f(x')| \leq 2\|f\|_\infty \leq \|f\|_\infty c\exp(vt/n - r).$$

$\qquad\square$

We can now prove Theorem 3.3.

*Proof of Theorem 3.3.* It is clear that $\mathrm{LP}^{\max}_{\Lambda,\mathrm{MC}}(f - \mu(f)) = \mathrm{LP}^{\max}_{\Lambda,\mathrm{MC}}(f) - \mu(f)$ so we may assume in our analysis that $\mu(f) = 0$. In this case by the fast mixing assumption, we know that $\|P^t f\|_\infty \leq \|f\|_\infty c_1 n(1 - c_2/n)^t$ for all $t \geq 0$.

Let $t_1 > 0$ to be fixed later and define $g = -\sum_{t=0}^{t_1-1} P^t f$. Note that

$$Pg = g + f - P^{t_1}f. \tag{21}$$

Given a function $h : \{-1, +1\}^V \to \mathbb{R}$, let $\tilde{h}$ be the function obtained from $h$ by fixing all the spins outside $\Lambda$ to some fixed value (say all $+1$) so that $\mathrm{supp}(\tilde{h}) \subseteq \Lambda$. By the constraints of the linear program, in particular the stationarity equalities in (17), we can then write for any feasible $\nu$:

$$\begin{aligned}
|\nu(f)| &= |\nu(f - (P\tilde{g} - \tilde{g}))| \\
&\leq \|f - (P\tilde{g} - \tilde{g})\|_\infty \\
&= \|Pg - g - (P\tilde{g} - \tilde{g}) + P^{t_1}f\|_\infty \\
&\leq \|(P - I)(g - \tilde{g})\|_\infty + \|P^{t_1}f\|_\infty
\end{aligned} \tag{22}$$

where the first inequality follows from the positivity condition of $\nu$, and the following equality follows from (21).

It remains to bound each term above individually. The first term can be bounded using Lemma 3.6. Indeed, using the latter we have, for any $x \in \{-1, +1\}^V$

$$|g(x) - \tilde{g}(x)| \leq \sum_{t=0}^{t_1-1} |P^t f(x) - \widetilde{P^t f}(x)| \leq t_1\|f\|_\infty ce^{vt_1/n-r}. \tag{23}$$

where $r = \mathrm{dist}(B, \Lambda^c)$. For the first term of (22), note that $(P-I)(g-\tilde{g})(x) = \mathbb{E}[(g-\tilde{g})(X_1) - (g-\tilde{g})(x)|X_0 = x] \leq 2\|g - \tilde{g}\|_\infty \leq 2t_1\|f\|_\infty ce^{vt_1/n-r}$. As a result,

$$\|(P - I)(g - \tilde{g})\|_\infty \leq 2t_1\|f\|_\infty ce^{vt_1/n-r}.$$

The second term of (22) can be bounded by the mixing time assumption. Putting things together we get

$$|\nu(f)| \leq 2t_1\|f\|_\infty ce^{vt_1/n-r} + \|f\|_\infty c_1 n(1 - c_2/n)^{t_1}.$$

Taking $t_1 = nr/2v$ gives

$$|\nu(f)| \leq C_1 n\|f\|_\infty e^{-C_2 r}$$

for some appropriate constants $C_1, C_2 > 0$, as desired. $\qquad\square$

## 4 Outlook

We considered certified algorithms for computing local expectation values of Gibbs distributions using linear programming relaxations. In practice, it is possible to strengthen the hierarchy using additional constraints, for example semidefinite programming (SDP) constraints such as the so-called *reflection positivity* constraints for the Ising model as done in Cho & Sun (2023).

Another variant of the hierarchy we considered in this paper is the following sequence of SDP relaxations (which can be interpreted as searching for sum-of-squares certificates) indexed by $\ell \in \mathbb{N}$:

$$\mathrm{SDP}_\ell = \Big\{ \nu \in \tilde{D}_{2\ell} : \begin{array}{ll} \nu(1) = 1 & \text{(Normalization)} \\ \nu(h^2) \geq 0 & \forall h : \deg(h) \leq \ell \quad \text{(Positivity)} \\ \nu(Pg) = \nu(g) & \forall g : \deg(g) \leq 2\ell - \Delta \quad \text{(Stationarity)} \end{array} \Big\}. \tag{24}$$

Here, $\tilde{D}_{2\ell}$ is the set of linear forms on the space of degree $2\ell$ polynomials in $(x_1, \ldots, x_n)$. Given a low-degree observable $f$, we let

$$\mathrm{SDP}_\ell^{\min}(f) = \min\{\nu(f) : \nu \in \mathrm{SDP}_\ell\}, \qquad \mathrm{SDP}_\ell^{\max}(f) = \max\{\nu(f) : \nu \in \mathrm{SDP}_\ell\}. \tag{25}$$

It is obvious to check that $\mathrm{SDP}_\ell^{\min}(f)$ and $\mathrm{SDP}_\ell^{\max}(f)$ are lower and upper bounds respectively on $\mu(f)$. An open question is to study the convergence rate of $\mathrm{SDP}_\ell^{\max}(f) - \mathrm{SDP}_\ell^{\min}(f)$ as $\ell \to \infty$ under appropriate conditions on the mixing time of the Markov chain.

Another interesting direction for future research is to use techniques from variational inference to improve the performance of the convex relaxations. Variational inference methods Wainwright et al. (2008) are alternatives to MCMC that can be used to compute expectation values of Gibbs distributions (among other things). Such methods are not certified (except for very specific instances), but have very good performance in practice in many settings of interest.

## Acknowledgements

HF would like to thank Oisín Faust and Shengding Sun for discussions that helped simplify the proof of Theorem 3.3, and Minjae Cho for discussions on the bootstrap. HF acknowledges funding from UK Research and Innovation (UKRI) under the UK government's Horizon Europe funding guarantee EP/X032051/1. OF acknowledges funding from the European Research Council (ERC Grant AlgoQIP, Agreement No. 851716).

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
