# OpenReview forum: "Convergence of linear programming hierarchies for Gibbs states of spin systems"
_TMLR — Accepted by TMLR_

### Review · Reviewer_6A1y · 2025-08-30

**Summary Of Contributions:**

This work focus on the theoretical convergence of two methods to obtain $\epsilon$-approximation of the expectation of local function under a Gibbs distribution of a spin system. The two methods are based on linear programs which allow to give lower and upper bounds on the integral. The main theoretical results are upper bounds on the difference of these bounds, which can be decreased to get an $\epsilon$-approximation.

**Audience:**

Yes

**Audience Explanation:**

This paper provides a theoretical result, justifying the use of 2 LP based methods to approximate integrals under a Gibbs distribution of a spin system, which have been used numerically in previous works. I am not very familiar with this literature, but it seems that it could interest some audience (but not sure for the TMLR's audience).

**Claims And Evidence:**

Yes

**Claims Explanation:**

This paper introduces the necessary background to understand the problem, and all the results are supported by proofs which seem correct.

**Requested Changes:**

The paper is very theoretical, and it is not clear to me what are the motivations to approximate this type of integral. I am aware that there is a literature that studies this problem, but for a TMLR paper, I believe that it would be important to better motivate the problem in the introduction.

It is also briefly mentioned in the Related work section that the two studied hierarchies have been used numerically in previous works. But it is not clear if these methods work well or not compared to other methods, which would then further motivate their theoretical study.

It is proved that the DLR bounds are sharper than MC bounds. Does it mean that the DLR method provides better results than the MC one?


Typos:
- Beginning of page 3: in the inequality, $\mathrm{LP}\_{\Lambda,\mathrm{DLR}}$ instead of $\mathrm{LP}_{\Lambda, \mathrm{MC}}$.
- Equation (13): misses DLR?

---

> ### Author Response · Authors · 2025-10-05
>
> We thank the reviewer for his review and comments. We address the comments one-by-one:
>
> * *Motivation* We have added a sentence in the introduction with references to indicate that the computation problem we consider is one of the typical problems in computational inference.
>
> * *Numerical comparison to other methods* We note that the works of Cho et al. have already compared numerically the methods we study here with Monte Carlo Markov Chain (MCMC). We added a sentence in the ``Related work'' section of the Introduction to say that numerically, certified methods lead to results comparable to MCMC away from criticality. We recall that the main advantage of the method studied in this paper is that it is certified i.e., we do not need an a priori analyses of the Markov chain to be confident about the results.
>
> * *Comparison between MC and DLR* For a fixed set $\Lambda$, the DLR bound is indeed sharper that the MC bound. But as the DLR bound has more constraints for a fixed $\Lambda$, it is not clear which linear program would give the best bound for a fixed computational cost.
>
> * *Typos* We have corrected the typos, thank you

---

### Review · Reviewer_rRjm · 2025-09-12

**Summary Of Contributions:**

This paper studies two families of linear programming (LP) hierarchies for approximating local expectation values under Gibbs distributions of spin systems. The first hierarchy is based on the DLR (spin-flip) equations, while the second is derived from local Markov chains (such as Glauber dynamics) having the Gibbs state as their stationary distribution. In both constructions, one selects a finite region $\Lambda \subseteq V$ (with $V$ the set of sites of the system) that contains the support $B$ of the observable of interest. The LP constraints are then imposed on $\Lambda$: spin-flip equalities for the DLR hierarchy, or local stationarity conditions for the Markov chain hierarchy. For any such finite $\Lambda$, the resulting LP provides valid upper and lower bounds on the observable. As $\Lambda$ grows, these bounds converge to the true expectation value. The authors show that under additional assumptions, this convergence is exponentially fast in the distance between $B$ and the complement $\Lambda^c$: spatial mixing of the Gibbs distribution suffices for the DLR-based hierarchy (Theorem~2.4), while a temporal mixing bound on the Markov chain (Eq.~(20)) suffices for the MC-based hierarchy (Theorem~3.3). On a $d$-dimensional grid and at sufficiently high temperature, the results imply that $\varepsilon$-accuracy can be obtained with LPs of quasipolynomial size in $n/\varepsilon$, specifically $\exp(O(\log^d(n/\varepsilon)))$.

**Audience:**

Yes

**Audience Explanation:**

The scope of the contribution is somewhat specialized: the focus is on spin systems and Gibbs measures, and the computational guarantees, while quasipolynomial, are unlikely to be practical at scale. Nevertheless, the perspective of using LP hierarchies for certified inference may be of interest to researchers in theoretical machine learning who are concerned with rigorous methods for inference in graphical models. For all these reasons, I am inclined to recommend this paper for publication in TMLR.

**Broader Impact Concerns:**

\begin{itemize}
\item The paper briefly mentions connections to variational inference and to sum-of-squares relaxations (e.g., in the related work section), but this point could be emphasized more strongly. Situating the LP hierarchies between variational inference (widely used but typically uncertified) and the sum-of-squares hierarchy (more powerful but less tractable) would help clarify where this contribution fits within contemporary ML theory.
\item How do the certified bounds established in the present paper extend to more general spin systems, such as the Potts model, hard-core models, and proper coloring models? It would be interesting to know if	 the more modern (and powerful) results by Lubetzky and Sly on general spin systems sampling (see, e.g., \textquotedblleft Cutoff for General Spin Systems with Arbitrary Boundary Conditions\textquotedblright, CPAM, 2014) could be employed in this direction.
\item {p.\ 1.} When deriving the $\delta^{-2}$ dependence, it may help to comment explicitly that this comes from a variance bound (e.g., Hoeffding’s inequality applied under approximate independence of samples).
\item {p.\ 2.} The hierarchy based on the Markov chain is claimed to be ``provably weaker.'' If I understand correctly, the justification comes from the discussion around equation (19), and this implication appears to rely on reversibility of the Markov chain. It would be useful to make this step transparent from the very the start, and refer to a later paragraph for the rigorous justification.
\item {p.\ 9.} In equation (22), the first line makes use of the third condition in (17) (the stationarity condition). Pointing this out would improve clarity.
\item {Typos.}
p.~2. ``corresponds'' $\to$ ``corresponding'';
p.~7. ``imposes'' $\to$ ``impose'';
p.~8. ``$X[B]$'' should read ``$X[B]$ be''.
\end{itemize}

**Claims And Evidence:**

Yes

**Claims Explanation:**

The paper is clearly written and technically correct. The analysis builds on classical results about spatial and temporal mixing of spin systems, combined with standard LP formulations. The main contribution is to present these ingredients in a unified framework and to highlight that LP relaxations provide {certified} bounds on observables. The mathematics is elementary but the exposition is careful and accessible.

**Requested Changes:**

\begin{itemize}
\item The paper briefly mentions connections to variational inference and to sum-of-squares relaxations (e.g., in the related work section), but this point could be emphasized more strongly. Situating the LP hierarchies between variational inference (widely used but typically uncertified) and the sum-of-squares hierarchy (more powerful but less tractable) would help clarify where this contribution fits within contemporary ML theory.
\item How do the certified bounds established in the present paper extend to more general spin systems, such as the Potts model, hard-core models, and proper coloring models? It would be interesting to know if	 the more modern (and powerful) results by Lubetzky and Sly on general spin systems sampling (see, e.g., \textquotedblleft Cutoff for General Spin Systems with Arbitrary Boundary Conditions\textquotedblright, CPAM, 2014) could be employed in this direction.
\item {p.\ 1.} When deriving the $\delta^{-2}$ dependence, it may help to comment explicitly that this comes from a variance bound (e.g., Hoeffding’s inequality applied under approximate independence of samples).
\item {p.\ 2.} The hierarchy based on the Markov chain is claimed to be ``provably weaker.'' If I understand correctly, the justification comes from the discussion around equation (19), and this implication appears to rely on reversibility of the Markov chain. It would be useful to make this step transparent from the very the start, and refer to a later paragraph for the rigorous justification.
\item {p.\ 9.} In equation (22), the first line makes use of the third condition in (17) (the stationarity condition). Pointing this out would improve clarity.
\item {Typos.}
p.~2. ``corresponds'' $\to$ ``corresponding'';
p.~7. ``imposes'' $\to$ ``impose'';
p.~8. ``$X[B]$'' should read ``$X[B]$ be''.
\end{itemize}

---

> ### Author Response · Authors · 2025-10-05
>
> We thank the reviewer for his review and comments. We address the comments one-by-one:
>
> * *Sum-of-squares and variational inference*: We added an outlook section at the end of the paper that should clarify the link to sum-of-squares relaxations as well as variational methods and provide some open questions in this direction.
>
> * *General spin systems*: Our results on the MC hierarchy are written for any Glauber dynamics where the number of possible configurations in each site is $2$. However, the exact same proof holds if the set of configurations is an arbitrary finite set $\Sigma$. However, we should keep in mind that the size of the linear program grows with the size of $\Sigma$: it is of order $|\Sigma|^{|\bar{\Lambda}|}$.
>
> * We added a footnote concerning the $\delta^{-2}$ dependence in the introduction: ``The $1/\delta^2$ factor comes from taking independent samples and ensuring the variance is $O(\delta^2)$.''
>
> * We modified the sentence in the introduction about the relation between DLR and MC (page 2) to reference Remark 3.2 and highlight the reversibility condition. We also introduced small modifications in Remark 3.2 to make it clearer.
>
> * We added to the sentence before equation (22): ``in particular the stationarity equalities in~(17)''
>
> * Typos: we corrected all of them, thank you.

---

### Review · Reviewer_pptv · 2025-09-24

**Summary Of Contributions:**

The authors investigate two hierarchical linear programming (LP) relaxations for computing the expectation of a local function \$ f \$ under the Gibbs distribution of a spin system:

1. **DLR-based Hierarchy:**
The first approach leverages the Dobrushin-Lanford-Ruelle (DLR) equations, resulting in LPs of size \$ 2^{\Theta(|\Lambda|)} \$, where \$ B \subseteq \Lambda \subseteq V \$. Here, \$ V \$ is the set of nodes in the graph, and \$ B \$ contains the nodes on which the function \$ f \$ depends.
2. **Glauber Dynamics Hierarchy:**
The second approach constructs a hierarchy grounded in Glauber dynamics, again leading to LPs of order \$ 2^{\Theta(|\Lambda|)} \$.

The central contribution of the paper is the development of methods that enable the computation of the expectation of local functions, with provable convergence guarantees.

**Audience:**

Yes

**Audience Explanation:**

I am not an expert in the area and feel that the topic may not be very relevant to a general ML audience. It is mostly relevant to researchers in theoretical physics. However, I do expect a potential overlap with some ML researchers (especially with those who work with Ising models).

**Broader Impact Concerns:**

No concerns.

**Claims And Evidence:**

Yes

**Claims Explanation:**

The authors formally propose their methods and provide convergence guarantees in Sections 2 and 3.

**Requested Changes:**

The paper is generally well written, and the proposed methods are underpinned by solid formal foundations and convergence guarantees. The novelty primarily lies in the introduction of these new LP hierarchies. The proofs draw carefully on results from prior literature.

A notable area for improvement could be the inclusion of empirical comparisons with MCMC-based baseline methods, which would significantly enhance the impact and practical relevance of the study.

---

> ### Author Response · Authors · 2025-10-05
>
> We thank the reviewer for his review and comments.
> We note that the works of Cho et al. have already compared numerically the methods we study here with Monte Carlo Markov Chain (MCMC). We added a sentence in the ``Related work'' section of the Introduction to say that numerically, certified methods lead to results comparable to MCMC away from criticality. We recall that the main advantage of the method studied in the present paper is that it is certified i.e., we do not need an a priori analyses of the Markov chain to be confident about the results.

---

### Decision · Action_Editor_dJ1w · 2025-10-29

**Recommendation:** Accept as is

**Audience:**

Yes

**Audience Explanation:**

The paper has a strong CS orientation but there clearly is a sub-community in the TMLR audience for that kind of theoretical works.

**Claims And Evidence:**

Yes

**Claims Explanation:**

This paper makes a strong theoretical contribution to certified inference in spin systems. It analyzes two linear-programming hierarchies and proves fast convergence under natural and interpretable conditions—spatial mixing (decay of correlations) and rapid mixing of a Markov chain that preserves the Gibbs state. The problem setting is timely, the assumptions are clearly stated, and the guarantees are technically solid and practically meaningful for regimes such as the high-temperature Ising model. The manuscript is clearly written and, per the latest revision, the authors have addressed reviewer feedback with added clarifications and an outlook section, improving readability and context. Overall, the combination of precise assumptions and clean convergence guarantees makes this a valuable addition to the literature on graphical models. I recommend acceptance.